# The Potential of Co-Fermentation with *Pichia kluyveri* and *Saccharomyces cerevisiae* for the Production of Low-Alcohol Craft Beer

**DOI:** 10.3390/foods13233794

**Published:** 2024-11-26

**Authors:** Ping-Hsiu Huang, Yung-Chi Lin, Yu-Wen Lin, You-Wei Zhang, Da-Wei Huang

**Affiliations:** 1School of Food, Jiangsu Food and Pharmaceutical Science College, No.4, Meicheng Rd., Higher Education Park, Huai’an 223003, China; hugh0530@gmail.com (P.-H.H.); garfield201507@126.com (Y.-W.Z.); 2Department of Biotechnology and Food Technology, Southern Taiwan University of Science and Technology, No.1, Nantai St., Yungkang Dist., Tainan 710301, Taiwan; mb0h0105@stust.edu.tw; 3Department of Food Science, Nutrition, and Nutraceutical Biotechnology, Shih Chien University, No.70, Dazhi St., Zhongshan Dist., Taipei 104336, Taiwan; ywlin@g2.usc.edu.tw

**Keywords:** fermentation kinetics, isoamyl acetate, phenyl ethyl acetate, quality indicators

## Abstract

The potential health impacts of moderate alcohol consumption have long been debated. The COVID-19 pandemic has heightened public awareness of health concerns, creating a clear market opportunity for low-alcohol craft beer development. This study investigated the possibility of low-alcohol craft beer by co-fermentation with different ratios of *Pichia kluyveri* (*P. kluyveri*) and *Saccharomyces cerevisiae* (SC) according to the established quality indexes. Specifically, this study was conducted to identify the low-alcohol craft beer quality by fermentation kinetics, growth kinetics, apparent attenuation (AA), real attenuation (RA), residual sugar content, alcohol by volume (ABV), and volatile organic compounds. This study demonstrated that the co-fermentation of SC and *P. kluyveri* in a 1:10 ratio produced an ABV of 2.98% (*v*/*v*). In addition, high concentrations of isoamyl acetate and phenyl ethyl acetate revealed banana, rose, apple, and honey flavors, respectively. Overall, this study revealed that the fermentation of *P. kluyveri* and SC by co-fermentation and the fermentation process by adjusting the yeast composition developed a craft beer with low alcohol content and rich aroma while establishing the quality indicators.

## 1. Introduction

Fermented foods have been integral to various traditional diets and cultures, likely stemming from the ancient domestication of microorganisms, potentially even before the domestication of plants and animals [1,2]. However, with advances in biotechnology and the development of genetically modified strains for use in fermented foods, there is a growing need for comprehensive regulatory standards to ensure food safety [3,4]. Public skepticism toward genetically modified foods and challenges in maintaining stability or cell viability under industrial conditions further complicate the acceptance and use of these strains [5]. However, there has been evidence that the primary pathogenicity of certain microorganisms is linked to their ability to damage host cells via enzymes such as cytolytic peptide toxins, which are more prevalent in pathogenic species like *Candida albicans* compared with nonpathogenic ones [6,7]. Moreover, some clinically discovered *Saccharomyces cerevisiae* strains secrete higher levels of proteases and phospholipases than those typically used in the food industry [8]. While the potential risks of accidental microorganism consumption are generally low for healthy individuals, they pose a greater threat to immunocompromised populations, including those on immunosuppressive drugs, broad-spectrum antibiotics, or with weakened immune systems (e.g., cancer, acquired immunodeficiency syndrome (AIDS), and hospitalized patients) [9]. Commercially used *Saccharomyces* strains, particularly in the beer industry, have undergone limited genetic hybridization with wild populations, adapting rapidly to industrial niches with minimal contact with the natural environment [5,10]. Therefore, novel strains, unrecognized applications, or genetically modified strains must undergo a comprehensive food safety evaluation or obtain clinical approval before use. There is a growing demand to improve fermented food production processes through advanced techniques, such as co-fermentation or non-*Saccharomyces* yeast strains, especially in various global regions [7]. This emphasis aligns with the belief that fermented foods significantly ensure food safety and sustainability [2,4].

Beer is a globally popular beverage, enjoyed as a hobby and consumed for pleasure, socializing, business, celebration, or even to ease difficult moments, making it a significant part of modern life [7,11,12,13]. According to the Brewers Association (BA) in the United States, craft beer is defined by the following standards: annual production of 6 million barrels or less, ownership of at least 75% of the brewery by the brewer, and exclusive brewing with malt, hops, water, and yeast. Reportedly, this fermentation of malt wort by yeast to produce a liquid containing alcohol and sugar has historically been used as an alternative to drinking water to ensure a risk reduction of foodborne illnesses without clean drinking water sources [4,12,14]. Moreover, in addition to the nutritional basics, it has bioactive properties, similar to other fermented foods, which provide specific health benefits [4,14]. Specifically, it can serve as a vehicle for delivering live microbes to the digestive tract or act as a prebiotic that offers selectively utilized substrates to host intestinal microorganisms, including oligosaccharides, polyphenolic compounds, and β-glucan present in the materials [4,14]. It is worth mentioning that these benefits depend on the recommended daily intake of guidelines suggested by countries or regions, accompanied by rational and moderate practice. However, it should be emphasized that alcohol consumption has been reported to be the 7th leading cause of death and disability-adjusted life-years at all ages [11,15]. Even low-to-moderate consumption of alcohol contributes to health issues, whereas complete abstinence from alcoholic beverages can minimize all the health risks [15]. Notwithstanding the support provided by some research regarding the health-promoting benefits of moderate alcohol consumption, the emergence of non-alcoholic beer has also been influenced by evolving health awareness and shifting attitudes toward alcohol [11,16,17,18]. However, the absence of ethanol alters certain sensory properties of the beer, which, depending on the method of ethanol removal, include possible manifestations of undesired flavors, caramelization, loss of volatile compounds, and color changes [11]. There have been indications that ethanol generation could be restricted using biological approaches during fermentation processes or by eradicating malt-originated aldehydes through cold-contact fermentation [19,20]. Interestingly, there have been several reports considering the incorporation of probiotics (lactic acid bacteria) into the beer process [14,21,22,23,24], with the expectation of conferring health-promoting benefits (anti-obesity, immunomodulation, anticancer, antidiabetic effects, etc.) to the beer in the same manner as probiotics [14,25,26,27], namely, by enhancing bioactive substance contents through the metabolic transformation of microorganisms [23,28]. The diversification of beer products is primarily driven by introducing innovative, realistic, and creative specialty beers [11]. This encompasses a range of novel flavors, low-calorie, non-alcoholic, and low-alcoholic beer (NABLAB) offerings [1,7,11,29]. Specifically, in the EU, “low-alcohol beers” with alcohol content from 0.5 to 1.2% *v*/*v* can be separated from non-alcoholic beers; in the USA, “non-alcoholic beers” is limited to 0.5% alcohol *v*/*v*; and in countries where religion is forbidden, the alcohol content of drinks should be limited to 0.05% *v*/*v* or less [13]. The beer brewed in our study has an alcohol content of 2.98%, which is lower than the typical alcohol content of commercial beers (4–5%). Therefore, low-alcohol beer is defined as beer with an alcohol content lower than that of commercial beers. Despite the popularity of craft beer worldwide in the last decades due to its exceptional sensory properties (added spices, herbs, flowers, and fruits), the need for more stringent quality control and reproducibility of the taste profiles commonly applied in its preparation deserves attention [30,31,32]. It is worth noting that research has demonstrated that incorporating various fruit juices in craft beer brewing to enhance fruit aroma has also increased ethanol content [33]. Beyond enhancing beer sensory profile by engendering novel attributes, such as varied tastes and flavors, fruits have been widely acknowledged as essential reservoirs of polyphenols and antioxidants, conferring significant functionality that facilitates health benefits [34]. It has also been reported that yeast strains isolated from traditionally fermented foods and 28 aroma-producing non-*Saccharomyces* yeast strains were selected for beer brewing, and many essential aroma compounds different from *Saccharomyces* were identified [35]. Specifically, these aromas included 2,6-nonadienal, esters (isoamyl ester, ethyl acetate, butyl ester, ethyl propionate, phenyl ethyl acetate, ethyl butyrate, and ethyl laurate), phenyl ethanol, 1-pentanol, acetic acid, and 3-methyl-4-heptanone, etc. [35]. The *P. kluyveri* has been applied to beer brewing under the above background while licensed with Generally Recognized As Safe (GRAS, GRN No. 938), albeit the published literature has been limited [29]. Remarkably, the low ethanol production and high acetate production properties have been used to improve the aroma quality of various alcoholic beverages while reinforcing the chocolate aroma of fermented cocoa and the coffee-related aroma of fermented green coffee beans [36,37,38,39]. Furthermore, these attributes have been harnessed in the hydrolysis of coffee grounds to yield chlorogenic and caffeic acid derivatives [38]. Moreover, businesses, scientists, and other stakeholders focus on nonconventional yeasts, identifying species and strains offering low ethanol yields and obtaining novel aroma profiles to develop suitable products that satisfy consumer demands (tastes, health, etc.) and expectations [40,41]. The craft beer revolution effectively capitalizes on emerging market opportunities, while industrially manufactured lagers maintain dominance in the global beer market [42]. Yet, according to a report, craft brewers can potentially surpass the current market share held by industrial beer brands, while these brands are presently taking strategic measures to offset any potential damage [43]. Consequently, the more pragmatic approach aligns with the market’s expectations for product advancement, such as NABLAB [13]. Therefore, this study aimed to develop a craft beer with low alcohol content (compared with the control group or traditional methods) and rich aroma via fermentation, namely, by modifying the microbial composition based on the co-fermentation of *S. cerevisiae* and *P. kluyveri*.

## 2. Materials and Methods

### 2.1. Materials

Malt (CHÂTEAU Vienna^®^, Europe brewery convention (EBC) color 4–7, the harvest year 2021) was purchased from La Malterie du Ch teau SA (Castle Malting) Malting Plant (Beloeil, Belgium). Two hops (*Humulus lupulus* L.) used in this study contained the Citra^®^ hops (α-acids 13.9%) purchased from Breeding Co., LLC. (Aurora, ON, USA), and the Czech Saaz hops (α-acids 4.1%) were purchased from Charles Faram & Co., Ltd. (Malvern, UK). SafAle™ US-05 Ale Yeast (*S. cerevisiae*) was purchased from Fermentis (Marcq-en-Barœul, France). *P. kluyveri* (BCRC 22537) strain was purchased from the Bioresource Collection and Research Center (BCRC, Hsinchu, Taiwan). Unless otherwise specified, all chemicals were purchased from Sigma-Aldrich^®^ (Merck KGaA, Darmstadt, Germany).

### 2.2. Examined Different Ratios of Yeast Strains of Fermentation Conditions

This study used the *P. kluyveri* strain that was activated based on the standard operating protocols (SOP) provided by the BCRC. Briefly, dried bacterial clumps or those adhering to the tube walls were lysed using 0.5 mL of yeast extract peptone dextrose (YPD) medium. Subsequently, 0.1 mL of the suspension was pipetted onto the YPD agar plate, evenly spread using the streak plate method, and incubated. The remaining suspension above was incubated by adding 10 mL of YPD medium for submerged fermentation. Both culture methods of *P. kluyveri* employed were assayed for activities and performance. Following activation, the procedure was carried out according to the method of François et al. [44] whereby the cultured *P. kluyveri* was incubated to an absorbance value of 0.5 (average bacterial count of 10^6^ cells/mL) at an absorbance wavelength of 580 nm. Next, commercial yeast (activated according to the manufacturer’s recommendations) was combined with the activated *P. kluyveri* broth in 1:1, 1:10, and 1:20 (*v*/*v*) ratios for co-fermentation. Afterward, 1 mL of different ratios (including two groups of the single strain of US-05 and *P. kluyveri*) broths were added to 25 mL of wort and incubated at 25 °C for 24 h. Next, 6 mL of each group broth was mixed with 300 mL of malt juice and incubated at 18 °C for another 24 h (activation) in conical flasks equipped with an airlock. Notably, the unused *P. kluyveri* broth was mixed with 30% glycerol at a 1:1 (*v*/*v*) ratio in sterile cryotubes and stored at −80 °C (at least one month).

### 2.3. Fermentation Condition Analysis

#### 2.3.1. Fermentation and Growth Kinetics Measurement

The fermentation and growth kinetics measurements were determined by the method of Canonico et al. [45] where carbon dioxide was produced and lost during the fermentation process. Therefore, the total weight of the flask for fermentation was measured every 24 h to calculate the weight of carbon dioxide lost, while the end point of fermentation was reached when the weight remained constant for three days. In addition, the growth kinetics were performed by sampling broth every 24 h of fermentation and counting colonies on a YPD agar medium. Namely, the count of yeast growth at each time point was expressed in log colony-forming unit (CFU)/mL.

#### 2.3.2. Apparent Attenuation (AA) Measurement

The original gravity (OG) of the initial wort and the fermented beer’s final gravity (FG) were determined separately by using a wine-specific gravity tester (Matsuhaku FMS-120Plato, Group Prospers Enterprise Co., Ltd., Taichung, Taiwan). Then, the BrewCalcs Brewing Calculator [46] was used to calculate the AA, expressed in percentages (%).
(1)$$Apparent\;Attenuation\;(AA,\%)=\frac{Original\;gravity\;(OG)-Final\;gravity\;(FG)}{OG-1}\times 100$$

#### 2.3.3. Determination of Sugar Residue Content Post-Fermentation

The residual sugar content of the samples collected after fermentation were determined using the Megazyme^®^ Maltose/Sucrose/D-Glucose assay kit (K-MASUG, Neogen Co., Lansing, MI, USA). The operation was carried out according to the SOP [47].

#### 2.3.4. Determination of Alcohol by Volume (ABV)

The OG and FG values were determined as described in Section 2.3.2 above, and the same equation was employed for the online calculation of ABV, which is expressed as a volume percentage.
(2)$$Alcohol\;by\;volume\;(ABV,\%)=\left(Original\;gravity\;(OG)-Final\;gravitie\;(FG)\right)\times 131.25$$

### 2.4. Craft Beer Brewing

Brewing operations and conditions for craft beer were based on the approaches described by Palombi et al. [48] and Peter et al. [49], with some modifications. Briefly, the malts were ground and mixed with 50 °C reverse osmosis water in a ratio of 1:4 (*w*/*v*) for saccharification in the Braumeister (20 L, Speidel Tank- und Behälterbau GmbH, Ofterdingen, Germany). The specific conditions were 52 °C for 1 min, 63 °C for 20 min, 68 °C for 25 min, and 73 °C for 25 min. Following saccharification, the juice was filtered to remove the malt, and the wort was boiled, followed by maintaining the post-boil temperature for 70 min. It is worth mentioning that 15 g of Citra^®^ hop was added upon boiling, and then 15 g of Czech Saaz hop was added at 45 min of cooking until the boiling was completed. Next, the wort was cooled to 25 °C and pre-activated yeast was added with (three groups of *S. cerevisiae*, *P. kluyveri*, and the above two strains at a ratio of 1:10) at 18 °C for five days of primary fermentation.

Next, the fermentation broths were bottled in 330 mL brown glass vials and fermented for 14 days at 4 °C. Samples were randomly collected at 7-day intervals during fermentation (0, 7, and 14 days) and storage (7, 14, 21, and 28 days) at 4 °C for quality indicator analyses.

#### 2.4.1. Determination of Free Amino Nitrogen (FAN) in Sweetened Wort

Upon completion of malt saccharification, the samples were collected and assayed for free amino nitrogen (FAN) using the assay kit (MAK449, Sigma-Aldrich, Merck KGaA, Darmstadt, Germany), following the standard operating procedures provided by the manufacturer. The following equation calculated the FAN content in the saccharified wort:(3)$$Free\;Amino\;Nitrogen\left(as\;Glycine;\;mM\right)=\frac{{ABS}_{Sample}-{ABS}_{Bla}}{Slope{mM}^{-1}}\times DF$$
where

ABS_sample_ represents the absorbance of the sample at 575 nm.

ABS_Bla_ represents the absorbance of the blank at 575 nm.

$Slope{mM}^{-1}$ is the slope of the standard curve.

DF means the dilution factor of the sample.

#### 2.4.2. Determination of Specific Gravity and Degrees Plato

The samples’ specific gravity (SG) and degrees Plato (°P) were determined using a wine-specific gravity tester (Matsuhaku FMS-120Plato, Group Prospers Enterprise Co., Ltd., Taichung, Taiwan). All measurements were conducted at room temperature with three replicates.

#### 2.4.3. Color Analysis (In EBC Units) and Appearance Evaluation

The beer color (Analytica EBC 9.6) determination was performed as described in EBC [50]. Briefly, the samples were degassed by stirring with a magnet and filtered through the 0.45 μm membrane. Then, the absorbance value of the filtrate was measured at 430 nm using a spectrophotometer (UV-1500PC, Macylab Instruments, Shanghai, China) with a 10 mm quartz cuvette, and the following equation calculated the EBC units.
Color (EBC units) = A × f × 25 (4)
where

A is the absorbance of the sample at 430 nm.

f is a dilution factor.

In addition, the samples were measured by a colorimeter (ZE-4000, Nippon Denshoku Industries Co., Ltd., Tokyo, Japan) to determine the *L**, *a**, and *b** values. The *L** value (0–100) indicates the perceived brightness, while the high value indicates the sample color is brighter. *a** positive value means red and negative value means green; *b** positive value means yellow and negative value means blue.

#### 2.4.4. Determination of Transmittance (%T) and pH

The determination of %T was based on the method described by Hughes [51] with minor modifications. The above samples were degassed and rewarmed to 25 °C, while %T of the sample was measured by a spectrophotometer (UV-1500PC, Macylab Instruments, Shanghai, China) at a wavelength of 650 nm, with distilled water serving as a control group. The sample’s %T was calculated using the following equation:Transmittance (%T) = (1 − Transmission value at 650 nm) × 100 (5)

In addition, the above degassed samples were measured for pH using a pH meter (PPL-700PV, GOnDO Electronic Co., Ltd., Taipei, Taiwan).

#### 2.4.5. Determination of International Bitterness Unit (IBU)

Determination of the sample’s IBU was performed according to the method described in K. McGivney [52]. The sample (10 mL) was supplemented with 1 mL of hydrochloric acid and 20 mL of iso-octane. Then, the sample was mixed until it was uniform and shaken at room temperature for 5 min, followed by centrifugation at 3000× *g* for 15 min. Afterward, the supernatant was transferred to another tube and protected from light for 30 min, and the absorbance was measured at 275 nm. Finally, the IBU of the sample was calculated using the following formula:International Bitterness Unit (IBU) = Absorbance of the sample at 275 nm × 50 (6)

#### 2.4.6. Determination of Foam Stability

Foam stability was performed according to the method described by Hiralal et al. [53], with some modifications. Briefly, the craft beer for this study was poured into the glass (355 mL; height 150 mm × diameter 63 mm) until the liquid level was 2 cm from the mouth of the glass. The time from the initial foam to the disappearance of the sample after pouring into the glass was observed.

### 2.5. Sensory Evaluation

The sensory evaluation was based on the method (Analytica EBC 13.1–13.3) described by EBC [50], with slight modifications. This study was conducted using consumer-based sensory evaluation methods described by Lin et al. [54] and included a panel of 20 members, ages 18–30, who had experience in drinking and consumption. The volunteers were informed in detail about the purpose of the sensory evaluation and the composition of all the samples (all edible and without any food safety risk) and explicitly knew and agreed to be part of the sensory evaluation study before participating. The samples were evaluated on days 14, 21, 28, 35, and 42 of fermentation (and on days 0, 7, 14, 21, and 28 of storage). The samples were dispensed in 15 mL plastic transparent tasting cups. The evaluation items include color, bubbles, aroma, taste, bitterness, and overall acceptability. Each item was scored on a 5-point scale where a higher score indicates the best item. Specifically, the score ranges from 1 (disliked very much), 2 (disliked a little), 3 (did not like or disliked), 4 (liked a little) to 5 (liked a lot). During the evaluation, the room temperature was kept at 25 ± 2 °C. There was no noise, no talking, or discussing with each other. Every time the sample was tasted, the panel members rinsed their mouths at least twice with drinking water until the flavor was off, then tasted the following samples. All completed scoring questionnaires were used for statistical analyses.

### 2.6. Determination of Flavor Compounds in Beer by Gas Chromatography–Mass Spectrometry (GC/MS)

The beer flavor compounds were determined based on the method described by de Lima et al. [55], with slight modifications. This study placed 5 mL of each sample in a sample bottle. The internal standard cinnamaldehyde was added to the samples by dilution with hexane (1:100, *v*/*v*). Afterward, the samples were incubated in a water bath at 50 °C for 30 min. The gases above the samples were adsorbed using solid-phase microextraction (SPME) for analysis. This study was analyzed using GC/MS (QP2010 SE, SHIMADZU Co., Kyoto, Japan) under the following conditions: the column was an Rtx-5MS GC capillary column (Rtx-5MS 30 m, 0.25 mm internal diameter (ID), 0.25 μm (film thickness), Restek Co., Bellefonte, PA, USA); the injection volume was 5 μL; injection temperature 280 °C; 1:10 split mode; oven temperature 40 °C; and carrier gas He at a total flow rate of 1.11 mL/min. The MS ion source temperature was 260 °C, and the interface temperature was 280 °C. The warming program was 40 °C for 2 min, increased at 10 °C/min to 140 °C, then increased at 7 °C/min to 280 °C, and maintained for 3 min. All results obtained from the analyses were compared with the database NIST20, and the peaks with a similarity score of 85 or more were screened as the primary target for qualitative analysis.

### 2.7. Statistical Analysis

The data in this study were presented as mean ± standard deviation (SD), and all analyses were conducted in triplicates (*n* = 3). The statistical analyses used GraphPad Prism (Version 9, Dotmatics, Boston, MA, USA). Within-group differences were analyzed using one-way analysis of variance (ANOVA), while between-group comparisons were conducted using the *t*-test. A significance level of *p* < 0.05 indicated a statistically significant difference.

## 3. Results and Discussion

### 3.1. Effects of Different Yeast Strains and Co-Fermentation at Various Ratios

#### 3.1.1. Fermentation and Growth Kinetics

This study showed that the commercial yeast (*S. cerevisiae*) alone produced the best fermentation results, followed by fermentation with a 1:1 ratio of commercial yeast to *P. kluyveri*. In contrast, *P. kluyveri* alone exhibited the lowest fermentation performance (Figure 1A). This phenomenon can be attributed to the slower fermentation kinetics of *S. cerevisiae* both in co-fermentation and alone [56]. Canonico et al. [56] also reported that strains exhibiting lower fermentation kinetics in wort are advantageous for low-alcohol beer production. It is worth mentioning that except for the *P. kluyveri* group, all other groups in this study were in line with the previously reported development trends of different yeast strains, with remarkable viability starting to appear at 3–7 days of incubation [40]. In particular, *S. cerevisiae* in this study was proved to be the most stable commercial strain regarding fermentation and growth profiles, and a similar result was also reported by Pirone et al. [33]. However, co-fermenting different yeast strains may regulate fermentation attenuation, leading to lower alcohol content and facilitating the production of low-alcohol craft beer. Therefore, the 1:10 ratio co-fermentation group was selected for the follow-up study as the fermentation performance of the 1:1 group was comparable to that of *S. cerevisiae* alone.

Regarding growth kinetics, this study showed that the *P. kluyveri* strain alone exhibited the best growth performance (Figure 1B) (*p* < 0.05), followed by the groups with co-fermentation at different ratios. In contrast, the *S. cerevisiae* showed the worst growth performance. It was hypothesized that these phenomena were attributed to the yeast extract, peptone, and glucose used in the YPD agar medium formulation, leading to poorer growth on the plate [16]. Moreover, in this study, all yeast cell growth during the initial 0–48 h primarily occurred during this period, which is in agreement with the published research results [16].

#### 3.1.2. Alcohol by Volume (ABV)

Commercially available Lager beers typically have an ABV range of approximately 4 to 5%. This study showed that the *S. cerevisiae* group exhibited the highest ABV (3.67%), followed by the 1:1 co-fermentation group with different yeast ratios (Figure 1C), and then, the 1:20, 1:10, and *P. kluyveri* groups, respectively, which were significantly different from each other (*p* < 0.05). Therefore, considering the above indexes, this study was conducted by selecting a blend of different yeasts for co-fermentation at a ratio of 1:10. Therefore, this study aimed to develop a craft beer with a lower alcohol content (compared with the control group or traditional methods) while ensuring satisfactory performance indicators.

#### 3.1.3. Apparent Attenuation (AA)

This study showed that the single strain group *S. cerevisiae* exhibited an AA of 51.14% upon completion of the primary fermentation (Table 1), which was a significant difference compared with others (*p* < 0.05). Conversely, the AA of the other single-strain *P. kluyveri* group was only 8.20%. Moreover, the AA of the three co-fermentation groups ranged from 45.93 to 47.87%. This phenomenon indicates that the co-fermentation of different yeasts contributes, to a certain degree, to the improved attenuation of *P. kluyveri*. Apart from having the ability to develop low-alcohol craft beers, it also prevents the beer from having an excess of fermentable residual sugars, avoiding undesirable mouthfeel and flavor.

#### 3.1.4. Residual Fermented Sugar Content and pH Value

This study showed that the *S. cerevisiae* group had the lowest residual content of maltose, sucrose, and D-glucose, while the 1:1 co-fermentation with different yeast strains ranked second (Table 1). Specifically, all these above groups could effectively utilize glucose and sucrose compared with the *P. kluyveri* group, whereas the residual content of maltose was less in the *S. cerevisiae* and 1:1 co-fermentation group. This phenomenon was attributed to the *P. kluyveri* only using glucose as a source of nutrition during the fermentation process [57]. *P. kluyveri* has been reported to be able to ingest only glucose and leave fructose behind during the brewing process, and although it can survive continued fermentation in 4–5% (*v*/*v*) ethanol, its fermentation capacity is insufficient to produce approximately 3.2% (*v*/*v*) ethanol [58]. In addition, Miguel et al. [29] reported three *P. kluyveri* strains were unavailable to metabolize sucrose and maltose in the beer-related medium with a preference for glucose as the nutrient source, compared with fructose. It also implies that except for *S. cerevisiae* and the 1:1 co-fermentation group, no other groups consumed significant amounts of maltose. Namely, these groups without consumption of malt will be particularly attractive for manufacturing low-alcohol beers [41]. The yeast strains used to brew beer have been reported to be capable of using a wide range of carbohydrates, such as glucose, fructose, maltose, galactose, raffinose, sucrose, and maltotriose [59]. The consumption of sugars is initiated by monosaccharides (glucose and fructose), followed by maltose and trisaccharides [14]. Sucrose is subsequently hydrolyzed to yield glucose and fructose [59]. Moreover, the wort has been reported to contain 20 to 30% nonfermentable dextrins apart from glucose, fructose, sucrose, maltose, and maltotriose [14,60], while dextrins contribute to beer body and drinkability [60]. Notably, these residual sugars provide excellent mouthfeel and sweetness upon beer fermentation [14]. In addition, either the wild yeast variant or the metamorphic indicator yeast *S. cerevisiae* var. diastaticus have been reported to carry STA1 genes, which code for the hydrolysis of dextrins, maltotriose, and other monosaccharides, as well as the breakdown of glucose from the nonreducing ends of oligosaccharides [3,14]. Moreover, in a metagenomic survey of the microbial composition of commercial beers, brewer’s yeast was found to be the most prevalent species though lower levels of wild yeasts were also detected (24 species observed in a single brew). Some beer samples contained more than ten different fungal species [61]. It has been hypothesized that the potential introduction of *S. cerevisiae* may have resulted from contamination during the manufacturing or sampling [7,61]. This suggests that extra yeasts beyond the conventionally acknowledged *S. cerevisiae* may have also been implicated, potentially exerting a larger-than-anticipated influence on beer production [7]. Therefore, it also implies the possibility of solving the issues of these sugar residues by using more natural methods such as co-fermentation (with nonconventional yeasts) combined with multi-stage fermentation, resulting in higher alcohol beers [62]. In contrast, this characteristic of impaired maltose fermentability could be considered for producing non-alcoholic beers [63].

Regarding pH, the five groups in this study were found to have pH values ranging from 4.52 to 4.81 upon primary fermentation with different yeast ratios. This was mainly attributed to the decrease in pH caused by the metabolism of the yeast, which produces carbon dioxide [28]. However, it has been reported that the inhibitory impact of a low pH environment on yeast carbohydrate consumption and metabolism is interrelated [64]. This study’s pH results align with the reported pH values of 3.5–5.5 for craft beers [14,44].

### 3.2. Effects of Co-Fermentation on the Quality Indicators of Craft Beers

#### 3.2.1. Free Amino Nitrogen (FAN) Content

This study showed that the FAN contents of *S. cerevisiae* and 1:10 groups were satisfactory (Table 1) significantly different (*p* < 0.05) compared with the *P. kluyveri* group. In addition, it has been documented that the FAN content of wort experiences a gradual decrease following fermentation [65]. *S. cerevisiae* has also been reported to uptake hydrophobic peptides from malt proteins to produce unique flavor compounds, namely, higher alcohols, organic acids, and esters [14,59]. However, the storage period post-fermentation does not yield any discernible variance in FAN levels [65], while the observed trend parallels the findings of this study. Moreover, it was reported that the amino acids of the three *P. kluyveri* strains were released in substantial amounts during the 48 to 144 h of fermentation at 20 °C in a synthetic wort medium [29]. At the same time, methionine was the most consumed amino acid regarding depletion. The preferences of different strains for amino acid consumption were consistent despite differences in their release and uptake of the amino acids [29]. In contrast, the FAN levels in this study’s *P. kluyveri* group were much higher than those in the *S. cerevisiae* and 1:10 co-fermentation groups. Thus, the difference between the literature and the above might be due to the variation in the composition of amino acids in the culture medium. Therefore, this study’s detailed distribution of amino acids should be identified and clarified in the future.

#### 3.2.2. Specific Gravity and Degrees Plato

In the winemaking process, an SG reading below 1.000 signifies the completion of fermentation facilitated by *S. cerevisiae* [66]. This study showed that SG and degrees Plato decreased as fermentation time increased in all groups (Figure 2A,B). Specifically, the groups exhibited pronounced changes on the third day of primary fermentation, with the SG and degrees Plato of the *S. cerevisiae* group decreasing significantly, followed by the 1:10 group, while the *P. kluyveri* group’s decreasing trend plateaued on the second day of primary fermentation. However, the three beers’ final (SG) and degrees Plato ranged from 1.033 to 1.049 and from 8.20 to 12.70 °P.

In terms of the bottle fermentation stage (0–14 days) followed by 28 days of storage stage (namely, 15–42 days of fermentation period), the SG and degrees Plato in all groups gradually plateaued at 14 days of bottle fermentation (Figure 2C,D). However, there remained a consistent and significant difference (*p* < 0.05) between all groups, in line with the above trends. It was attributed primarily to the nonavailability of an extra carbon source (such as glucose) at this stage, which occurred as the yeast completed the fermentation by consuming the sugar in the wort during the primary fermentation. Therefore, the final SG and degrees Plato for the three groups ranged between 1.016 and 1.049 and 4.07 and 12.17 °P, respectively.

#### 3.2.3. Color (EBC Units)

The research demonstrates that color is pivotal in consumers’ acceptance of foods, rendering it one of the most crucial attributes [67,68]. However, consumers typically prefer visually appealing beer colors, encompassing a broad spectrum ranging from light to dark hues, including gold, yellow, pale straw, amber, copper, and brown to black [68,69,70]. This study showed that the EBC ranged from 19 to 21 for different yeasts and the 1:10 co-fermentation ratio (Figure 2E), but there were slight significant differences (*p* < 0.05) between all groups. This also implied no variation in coloration depending on the fermentation strains. Interestingly, the color of beer comes mainly from the phenolic compounds in the malt or hops, which, apart from providing the beer with a variety of colors, contribute to the protection of light-sensitive elements, which, in turn, facilitate the preservation of beer [68,69]. Furthermore, the thermal treatment of the formulated ingredients leads to primary color changes in the beer through the Maillard reaction (MR) and caramelization of sugars and amino acids [68,69]. It has been reported that polyphenol oxidation is also implicated in these color changes [69].

#### 3.2.4. Appearance Color

This study showed that the *L** values of the three groups of craft beers increased (*p* < 0.05) with the time (bottling fermentation (0–14 days) and storage (21–42 days)) (Table 2). The order from brightest to darkest was as follows: *S. cerevisiae* group was the brightest, followed by the 1:10 co-fermentation group, and the *P. kluyveri* group was the darkest. Conversely, the *a** and *b** values were the highest in the *P. kluyveri* group; namely, the appearance of the colors was reddish and yellowish, while the *S. cerevisiae* and 1:10 co-fermentation groups were similar. Therefore, these phenomena can be attributed to the subtle effects of different yeast strains despite the influence of material (malt or hops) and heat treatment during manufacturing on MR production and polyphenol oxidation [68,69]. However, it should be noted that in this study, the observed variations primarily stem from the utilization of distinct yeast strains.

#### 3.2.5. Transmittance (%T) and pH Value

This study showed that the %T of each group increased with the duration of the bottling and storage process (Figure 2F). Specifically, on days 0–14 of bottle fermentation, the yeast remained suspended in the liquid due to the interruption of fermentation, leading to a low %T. However, there was a significant increase in %T for each group on day 21 of the storage process, namely, during fermentation, which matched the trend of SG and degrees Plato described above (as defined in Section 3.2.2). This also implied that the fermentation of the craft beers in this study was completed on day 14, involving the release and settling of yeast residues and secondary metabolites. Another possible explanation might be that the polyphenols of the wort interact with lipids and proteins during the fermentation process, forming insoluble sediments that gradually accumulate and, subsequently, impact beer turbidity [71]. In addition, it has been documented that turbidity, a characteristic resulting from the absence of filtration to remove all biomass in craft beer production, serves as a typical indicator of its quality [31].

Regarding pH, all groups in this study showed stabilized pH values (Figure 2G), except for the *P. kluyveri* group, which showed a slight decrease from 0 to 7 days of bottle fermentation, yet all groups were under the pH range of commercially available beers, 4–6. Among them, the *P. kluyveri* group was the highest (*p* < 0.05), followed by the pH of *S. cerevisiae* and 1:10 co-fermentation groups. It is worth mentioning that it was reported that the low pH of beer resulted in some negative flavors during storage [72]. In contrast, the high pH was detrimental to beer preservation.

#### 3.2.6. International Bitterness Units (IBU)

Typically, the IBU of beer ranges from 5 to 100, yet the human tongue can perceive the bitter flavor from 12 IBU, whereas no bitter taste can be perceived below 6 IBU. Specifically, bitterness has been defined as normal bitterness (≤20 IBU), medium bitterness (21–40), and very bitter (≥41 IBU) [73]. This study showed that the IBUs of the three groups of craft beers were not significantly different during bottle fermentation and storage (Figure 2H). Still, there were significant differences between groups (*p* < 0.05). Specifically, the *S. cerevisiae* and 1:10 co-fermentation groups had bitterness levels ranging from 29 to 36 IBUs, exhibiting medium bitterness in taste. However, the *P. kluyveri* group had a bitterness level of 50–60 IBU, comparable to the bitterness values of the commercially available Vienna lagers with EBC [50] Method 9.8. Moreover, the bitter flavor in beer primarily arises from the α-acids of hops, which include five similar structures: humulone, cohumulone, adhumulone, prehumulone, and posthumulone [74,75]. However, the α-acids change to the more soluble and bitter iso-α-acids (including isohumulone, isocohumulone, and isoadhumulone) by boiling, while hops provide the primary source of the multilayered flavor in beer [74,75]. Therefore, the hops used in this study, Citra, with 13.9% of α-acids, were of distinct bitterness and aroma, whereas Saaz, with 4.1% of α-acids, contributed primarily to aroma [73].

#### 3.2.7. Alcohol by Volume (ABV)

This study showed that the ABVs of the three groups were slightly increased with the period of fermentation and storage in vials (Figure 2I). The highest ABV was recorded in the *S. cerevisiae* (4.56%) group, followed by the 1:10 co-fermentation group (2.98%), and the lowest in the *P. kluyveri* group (0.44%) after the fermentation, which was significantly different from each other (*p* < 0.05). It is important to note that the *P. kluyveri* strain cannot metabolize maltose in the wort. However, it has been documented that the alcohol content from fermentation utilizing a beer-related medium directly correlates to the initial glucose concentration and the conversion rate of approximately 0.50 g alcohol/g glucose [29]. Another reported use of malt wort for fermentation with the *P. kluyveri* strain yielded an even lower alcohol content of 0.33% (*v*/*v*, no stirring) [63]. Furthermore, stirring led to a higher alcohol content of 0.67% (*v*/*v*), as opposed to 0.50% (*v*/*v*) in the absence of stirring [29].

#### 3.2.8. Foam Stability

Beer foam is a characteristic for evaluating beer quality, determined by the quality of malt and hops used in brewing and stability [76]. Specifically, the stability of beer foam is influenced by specific components, including proteins, FAN, various chain lengths of free fatty acids, and saturated and unsaturated fatty acids [75,77,78]. Moreover, other brewing-related vital variables, such as raw materials (malt, grains, and hop varieties), fermentation process, and storage, have been reported to be affected to a certain extent [48,79], extending even to the physicochemical and sensory properties of the final product [48,59]. Notably, these proteins in malt have been found to enhance yeast development and provide beer with roasted and smoked aromas, including, but not limited to, biscuits, honey, cinnamon, bread, chocolate, cocoa, coffee, etc. [75]. It has been reported that a regular beer foam head retention time should be at least 5 min [80]. This study revealed that both *S. cerevisiae* and 1:10 co-fermentation groups complied with a regular beer-required foam head retention time (Figure 2J). Reports indicate that using foam-boosting syrup for stabilization has historically been prevalent, primarily with barley malt, while using wheat and oats as adjuncts has been customary [32]; these approaches also indirectly contribute to developing diverse flavor profiles. Moreover, it has been suggested that changes in fermentation temperature and increased pH of the fermentation medium reduce foam head stability [53].

### 3.3. Effects of Co-Fermentation on the Sensory Evaluation and Volatile Organic Compounds of Craft Beers

This study showed that the commercially available beer was the best in terms of color, bubbles, aroma, taste, bitterness, or overall acceptability, while the 1:10 and *S. cerevisiae* groups were the following best, and the *P. kluyveri* group was the worst (Figure 3A–E). Notably, the panelists were unimpressed with the *P. kluyveri* group. They provided feedback on high bitterness, poor taste, and lack of air bubbles and foam, consistent with the results mentioned earlier in the quality indicator analyses. However, the 1:10 co-fermentation of *S. cerevisiae* groups and *P. kluyveri* yeast strains was the most preferred beer by the panelists in this study.

Beer has been reported to exhibit more than 800 volatile organic compounds, including esters, higher alcohols, organic acids, sulfur compounds, carbonyls, and short-chain fatty acids [59], depending on hop type (containing hydrocarbons (monoterpenes), oxygenates (terpene alcohols), and sulfur compounds), and addition time. In contrast, craft beers tend to have more complexity in terms of flavor compared with industrial beers [5]. In addition, the sugar and amino acid compositions, yeast species, and the manner of utilization affect the yield and sensory profile of the end product [5]. This study showed that isoamyl acetate, ethyl acetate, and phenyl ethyl acetate were relatively higher in the *P. kluyveri* group (Table 3), which agrees with the results reported by Holt et al. [81]. Notably, isoamyl acetate and ethyl acetate have been reported to generate a certain degree of liaison, ultimately resulting in a sweet, fruity bouquet [31,82]. However, the yeast strains used in this study were selected as a 1:10 blend of *S. cerevisiae* and *P. kluyveri* for craft beer brewing, yielding the second-highest levels of these ester flavor compounds compared with the *P. kluyveri* group. It is worth mentioning that isoamyl acetate, with a unique banana flavor [81], and the rose flavor of phenyl ethyl acetate showed a significant increase in the 1:10 co-fermentation group. It has been reported that brewing with *P. kudriavzevii* 4A produces a pale ale with sufficient bitterness and brightness while providing a fruity flavor [31]. Specifically, Nieto-Sarabia et al. [31] stated that isopropyl alcohol (alcoholic, wine, or sweet aroma), ethyl acetate (fruity or solvent flavor), and low concentrations of isopropyl acetate (banana flavor) were contained. In addition, Methner et al. [16] reported that the beer brewed by *P. kluyveri* (isoamyl acetate 3.43 mg/L) was identified in the sensory evaluation by only a few panelists as exhibiting flavors of banana, cool mints, and solvents. It has also been reported that sweet and tart apple aromas provided the overall flavor balance of the beer [48,83]. Despite previous reports indicating the formation of acetaldehyde and aldehydes through alcohol oxidation during the fermentation and storage of beer, as well as the generation of Strecker aldehyde during pasteurization, this study did not yield any detection of aldehydes [31,84].

Moreover, the abundance of aromatic compounds in beer has been reported to be produced by yeast via the catabolic Erhlich pathway from amino acid catabolism or carbohydrate metabolism [41,59,85]. Specifically, the production of esters and volatile compounds can be influenced by various factors, including species-specific relationships, different fermentation conditions (adjuvant, aeration, and sequencing), and different yeast strains [41,53,86]. Despite ethyl acetate contributing to the fruity flavor of beer, it remains the most prominent compound in the beer ester profile but falls within the perception threshold (20–30 mg/L) considered to have a minor impact on beer taste [59,71]. Remarkably, it involves the threshold concentration of individual esters, which affects the beer flavor profile apart from the synergistic effects on the individual flavors [59,85]. Some studies suggest that these sensory evaluations require analyses of homogeneous groups of individuals based on consumers’ personal preferences to avoid distorted or misleading results from using averages [71,87]. In addition, it has been reported that the use of the extensive sensory database provided by some famous beer society platforms, combined with analysis by specialist sensory teams to train a set of machine learning models for profiling and understanding the performance of complex flavors, can help identify specific compounds as potential drivers of beer flavor and appreciation [12]. However, for beverages or food products with strong hobby attributes, it is crucial to prioritize and focus on the specific consumer group the product intends to target. Otherwise, the composition and discussion of the best-flavored foods would be never-ending, regardless of the definition, which depends principally on the market value as a driving factor [88].

Altogether, it has been suggested that *P. kluyveri* should not be used in isolation, namely, that different *P. kluyveri* strains and fermentation conditions can modulate fermented beverages’ flavor and aroma [89]. Similar trends were observed in this study. Therefore, this study’s 1:10 co-fermentation ratio of yeast strains showed potential for brewing low-alcohol, richly flavored craft beers compared with using these two yeast strains individually.

## 4. Conclusions

This study showed that the co-fermentation of *S. cerevisiae* and *P. kluyveri* in the ratio of 1:10 exhibited satisfactory fermentation performance and attained a relatively acceptable level of acceptance by the consumer-type panelists. Specifically, it had a similar taste, aroma, or bitterness as commercially available beers. Still, it contributed to the aroma due to incorporating *P. kluyveri*, namely, due to an increased phenyl ethyl acetate and isoamyl acetate and a lower alcohol content. However, this also implies that no additional operations were performed to remove alcohol, thus preserving the beer flavor and maintaining energy savings. In this study, the quality indicators for the physicochemical properties of the low-alcohol beer prepared by co-fermentation were as follows: initial SG ≥ 1.054; initial degrees Plato ≥ 13.8°P; ABV ≤ 3%; %T ≥ 80%; pH 4.3; foam head retention ≥ 5 min. However, the limitation of this study remains that the relationship between hydrocarbon, reducing sugar, and sensory properties has not been clarified. Moreover, several conditions could still be incorporated in co-fermentation and improve the optimization process, such as continuous stirring, aeration, staged fermentation, or other probiotics. Above all, this study provides information and potential opportunities to develop craft beer iterations by offering novel product attributes. Ultimately, through co-fermentation and the exploration of non-alcoholic beers, it will be possible to expand the health-promoting options of specialty beers (lower alcohol with rich bioactive substances) and to have the various sensory profiles desired by consumers, among other advantages, to maintain food safety and satisfaction.

## Figures and Tables

**Figure 1 foods-13-03794-f001:**
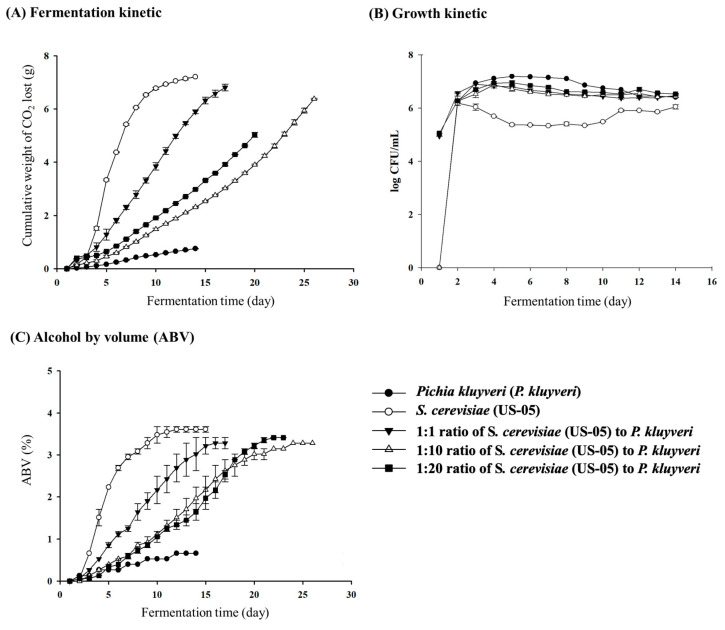
Effects of different yeast strains and co-fermentation ratios on (**A**) fermentation kinetics, (**B**) growth kinetics, and (**C**) alcohol by volume (ABV).

**Figure 2 foods-13-03794-f002:**
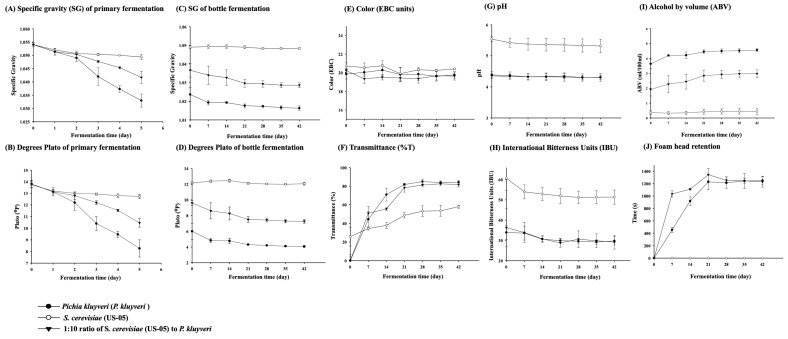
Effects of co-fermentation on the quality indicators contain (**A**) specific gravity (SG) of primary fermentation, (**B**) degrees Plato of primary fermentation, (**C**) SG of bottle fermentation, (**D**) degrees Plato of bottle fermentation, (**E**) color (EBC units), (**F**) transmittance (%T), (**G**) pH values, (**H**) international bitterness units (IBU), (**I**) alcohol by volume (ABV), and (**J**) foam head retention of craft beers.

**Figure 3 foods-13-03794-f003:**
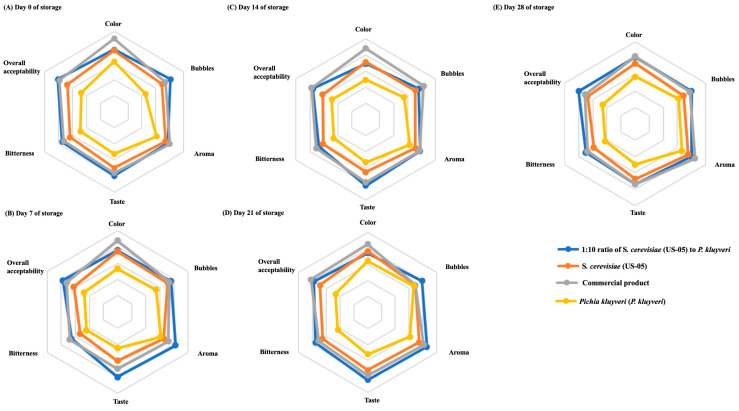
Effects of storage on sensory evaluations of co-fermented craft beers at (**A**) Day 0, (**B**) Day 7, (**C**) Day 14, (**D**) Day 21, and (**E**) Day 28.

**Table 1 foods-13-03794-t001:** Effects of co-fermentation yeast compositions on the quality indicators such as residual sugar content, pH, apparent attenuation (AA), and free amino nitrogen (FAN) content of craft beers.

Item		Wort	*Pichia kluyveri* (*P. kluyveri*)	*S. cerevisiae* (US-05)	1:1	1:10	1:20
Residual sugar content (g/L)	Glucose	2.60 ± 0.01 ^a^	0.20 ± 0.01 ^b^	0.02 ± 0.02 ^b^	0.03 ± 0.02 ^b^	-	-
	Sucrose	6.48 ± 0.26 ^a^	1.84 ± 0.07 ^b^	-	-	-	-
	Maltose	8.29 ± 0.56 ^a^	6.14 ± 2.56 ^a^	0.11 ± 0.05 ^d^	1.33 ± 0.01 ^c^	4.15 ± 0.10 ^b^	4.36 ± 0.11 ^b^
pH		5.47 ± 0.01 ^a^	4.81 ± 0.01 ^b^	4.54 ± 0.01 ^b^	4.66 ± 0.00 ^b^	4.56 ± 0.00 ^b^	4.52 ± 0.00 ^b^
Apparent attenuation (AA, %)		-	8.62 ± 0.00 ^a^	51.14 ± 2.86 ^b^	46.81 ± 1.19 ^c^	45.93 ± 2.07 ^b^	47.87 ± 2.19 ^b^
Free amino nitrogen (FAN, mg/L)		139.10 ± 8.15 ^a^	125.42 ± 1.16 ^a^	50.92 ± 1.15 ^b^	-	48.12 ± 2.31 ^b^	-

*S. cerevisiae* (US-05) was 1, while *P. kluyveri* were ratios of 1, 10, and 20, respectively. Different superscripted lowercase letters in the same row represent significant differences (*p* < 0.05).

**Table 2 foods-13-03794-t002:** Effects of co-fermentation on the appearance color (*L**, *a**, and *b** values) of craft beers.

Days	*S. cerevisiae* (US-05)			*Pichia kluyveri* (*P. kluyveri*)			1:10		
	*L**	*a**	*b**	*L**	*a**	*b**	*L**	*a**	*b**
0	14.52 ± 1.24 ^a^	5.87 ± 0.76 ^a^	20.16 ± 1.53 ^a^	37.28 ± 0.43 ^c^	15.58 ± 0.52 ^c^	44.39 ± 1.18 ^c^	20.84 ± 0.83 ^b^	9.01 ± 0.46 ^a^	27.29 ± 1.91 ^a^
7	54.40 ± 1.69 ^a^	13.77 ± 1.23 ^a^	46.75 ± 2.25 ^a^	37.94 ± 0.78 ^b^	16.66 ± 0.49 ^a^	45.01 ± 0.48 ^a^	44.47 ± 3.29 ^a^	14.70 ± 0.43 ^a^	44.32 ± 1.14 ^a^
14	65.00 ± 0.19 ^a^	12.30 ± 1.52 ^a^	45.71 ± 3.87 ^a^	39.94 ± 0.39 ^c^	16.91 ± 0.59 ^a^	45.53 ± 0.17 ^a^	59.66 ± 1.01 ^a^	14.13 ± 0.68 ^a^	44.50 ± 2.10 ^a^
21	69.97 ± 1.98 ^a^	11.94 ± 1.59 ^a^	47.87 ± 1.70 ^a^	45.58 ± 0.87 ^c^	16.79 ± 0.92 ^a^	46.10 ± 0.91 ^a^	64.59 ± 1.63 ^a^	14.01 ± 0.79 ^a^	45.17 ± 2.20 ^a^
28	74.51 ± 1.83 ^a^	11.04 ± 0.66 ^a^	48.20 ± 2.36 ^a^	47.77 ± 0.72 ^b^	16.91 ± 0.84 ^a^	47.44 ± 0.77 ^a^	70.61 ± 0.32 ^a^	12.37 ± 0.92 ^a^	45.47 ± 1.82 ^a^
35	76.31 ± 2.10 ^a^	11.76 ± 1.63 ^a^	47.73 ± 2.27 ^a^	48.77 ± 1.54 ^c^	16.83 ± 0.91 ^a^	47.65 ± 0.72 ^a^	74.70 ± 1.17 ^a^	12.51 ± 0.81 ^a^	45.50 ± 2.01 ^a^
42	77.42 ± 1.63 ^a^	11.91 ± 1.66 ^a^	48.75 ± 1.44 ^a^	50.89 ± 3.15 ^b^	16.71 ± 1.08 ^a^	48.58 ± 0.96 ^a^	75.52 ± 1.08 ^a^	13.99 ± 1.53 ^a^	46.47 ± 1.53 ^a^

*S. cerevisiae* (US-05) was 1, while *P. kluyveri* was a ratio of 10. Different superscripted lowercase letters in the same column represent significant differences (*p* < 0.05).

**Table 3 foods-13-03794-t003:** Effects of co-fermentation on the volatile organic compounds of craft beers.

Compound	*S. cerevisiae* (US-05)	*Pichia kluyveri* (*P. kluyveri*)	1:10
Cinnamaldehyde	1.00 ± 0.00	1.00 ± 0.00	1.00 ± 0.00
Ethanol	1.02 ± 1.18 ^a^		0.55 ± 0.49 ^b^
Ethyl Acetate	0.13 ± 0.02 ^c^	0.46 ± 0.05 ^a^	0.31 ± 0.06 ^b^
n-Hexane	6.85 ± 4.78 ^a^	1.92 ± 0.86 ^b^	6.08 ± 2.75 ^a^
Isoamyl alcohol	0.41 ± 0.15 ^a^		0.31 ± 0.10 ^b^
Isoamyl acetate		1.19 ± 0.43 ^a^	0.26 ± 0.15 ^b^
Ethyl hexanoate			0.23 ± 0.10
Phenylethyl Alcohol	0.72 ± 0.10 ^a^		0.17 ± 0.07 ^b^
Caprylic acid	0.63 ± 0.16 ^a^		0.24 ± 0.07 ^b^
Ethyl octanoate	0.79 ± 0.18 ^a^		0.84 ± 0.37 ^a^
(S)-2-Methylbutyl acetate		0.09 ± 0.03	
Furfuryl acetate		0.09 ± 0.04	
Linalool		0.04 ± 0.01	
Phenylethyl acetate	0.18 ± 0.01 ^b^	1.03 ± 0.25 ^a^	0.26 ± 0.08 ^b^
Neryl acetate		0.06 ± 0.02	
Humulene		0.04 ± 0.01	
Butylated Hydroxytoluene	0.72 ± 0.10 ^a^	0.41 ± 0.07 ^b^	0.36 ± 0.08 ^b^

*S. cerevisiae* (US-05) was 1, while *P. kluyveri* was a ratio of 10. Different superscripted lowercase letters in the same row represent significant differences (*p* < 0.05).

## Data Availability

Data included in the article and all the data supporting this study’s findings are available from the corresponding author upon reasonable request. The data are not publicly available due to privacy restrictions.

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
