# Peer review of "The Potential of Co-Fermentation with Pichia kluyveri and Saccharomyces cerevisiae for the Production of Low-Alcohol Craft Beer"

_foods, 2024, doi:10.3390/foods13233794_

Round 1
Reviewer 1 Report
Comments and Suggestions for Authors
I am very grateful to you for the invitation to review the manuscript foods-3319491 by Huang and coauthors, titled "Potential of co-fermentation with Pichia kluyveri and Saccharomyces cerevisiae for the production of low-alcohol craft beer". This study investigated the possibility of low-alcohol craft beer by co-fermentation with different ratios of Pichia kluyveri (PK) and Saccharomyces cerevisiae (SC) and then for established quality indexes. Specifically, this study was conducted to identify the low-alcohol craft beer quality by symbiotic fermentation kinetics, growth kinetics, apparent attenuation (AA), real attenuation (RA), residual sugar content, alcohol by volume (ABV), and volatile organic compounds. The work is interesting but needs adjustments to improve the quality of the material.
Comments:
- Line 16: Describe what low-alcohol beers are (concentration).
- Abstract: Please provide a detailed, step-by-step description of the research, including a more thorough explanation of the techniques and conditions used.
- Line 22: Is it possible to classify it as low-alcohol beer with this ethanol concentration? Specify with regard to relevant legislation.
- Abstract: Present the specific results more clearly. Insert numerical data related to the key findings of the study. Provide a clearer explanation of what is meant by "high."
- Line 27: Replace repeated keywords with alternative terms not mentioned in the title.
- Line 54: Change “regions. [7].” to “regions [7].”
- Line 63: Specify the health effects.
- Line 86: Further specify the low-alcohol category.
- Introduction: Better characterize the definition and market of craft beers.
- Materials and Methods: Check the adequacy of equations and ensure style standardization.
- Lines 188 and 190: Standardize units throughout the text (e.g., min, minutes, etc.).
- Line 219: Briefly explain the procedure and any modifications made.
- Line 264: Insert the protocol number of the relevant ethics committee approval.
- Lines 313-315: Provide a clearer explanation of microbial metabolism and the inhibitions that occur in co-culture.
- Line 360: Further explain the term "attenuation."
- Line 414: Clarify the classification of alcohol in relation to beer commercialization. Regardless of higher or lower concentration, products are classified as alcoholic beverages under the same legislation.
- Lines 484 and 489: Provide more explanations on color, including the reactions involved in color production in beers.
- No relevant discussion is presented under "Appearance color." Please expand this section.
- Line 541: In this case, would the beer be classified as non-alcoholic? Explain this better and describe the relevant legislation.
- Include an image of the produced beers.
Author Response
Reviewer #1: I am very grateful to you for the invitation to review the manuscript foods-3319491 by Huang and coauthors, titled "Potential of co-fermentation with Pichia kluyveri and Saccharomyces cerevisiae for the production of low-alcohol craft beer". This study investigated the possibility of low-alcohol craft beer by co-fermentation with different ratios of Pichia kluyveri (PK) and Saccharomyces cerevisiae (SC) and then for established quality indexes. Specifically, this study was conducted to identify the low-alcohol craft beer quality by symbiotic fermentation kinetics, growth kinetics, apparent attenuation (AA), real attenuation (RA), residual sugar content, alcohol by volume (ABV), and volatile organic compounds. The work is interesting but needs adjustments to improve the quality of the material.
Response: Thank you very much for your precious comment. All revisions in the manuscript are shown in yellow.
Comments:
- Comments 1: Describe what low-alcohol beers are (concentration).
Response 1: Thank you for your advice. The relevant information in the manuscript has been revised, and the details are as follows.
Lines 92-98: " Specifically, in the EU, “low-alcohol beers” with alcohol content from 0.5–1.2% v/v can be separated from non-alcoholic beers, while in the USA, “non-alcoholic beers” is limited to 0.5% alcohol v/v, and in countries where religion is forbidden, the alcohol content of drinks should be limited to 0.05% v/v or less [14]." The beer brewed in our study has an alcohol content of 2.98%, which is lower than the typical alcohol content of commercial beers (4-5%). Therefore, low-alcohol beer is defined as beer with an alcohol content lower than that of commercial beers
- Comments 2: Abstract: Please provide a detailed, step-by-step description of the research, including a more thorough explanation of the techniques and conditions used.
Response 2: Thank you very much for your precious comment. Considering the word limit and the loss of readability of an excessively long abstract, we have included a list of the essential techniques and the optimal conditions in the abstract. Detailed conditions and processes have also been presented in the "Materials and Methods" or "Discussion and Results" section.
- Comments 3: Line 22: Is it possible to classify it as low-alcohol beer with this ethanol concentration? Specify with regard to relevant legislation.
Response 3: Thank you for your kind suggestion. It has been revised in the manuscript (lines 92-98).
- Comments 4: Abstract: Present the specific results more clearly. Insert numerical data related to the key findings of the study. Provide a clearer explanation of what is meant by "high."
Response 4: Thank you for your kind suggestion. The research data results in the manuscript are considered significantly higher only when compared to the control group and exhibit statistical significance
- Comments 5: Line 27: Replace repeated keywords with alternative terms not mentioned in the title.
Response 5: Thank you for your kind suggestion. Specifically, we deleted the word "co-fermentation", which overlapped with the title.
- Comments 6: Line 54: Change “regions. [7].” to “regions [7].”
Response 6: Thank you for pointing out this error. It has been revised in the manuscript.
- Comments 7: Line 63: Specify the health effects.
Response 7: Thank you for your kind suggestion.
Lines 66-69: Specifically, it can serve as a vehicle for delivering live microbes to the digestive tract or act as a prebiotic that offers selectively utilized substrates to host intestinal microorganisms, including oligosaccharides, polyphenolic compounds, and β-glucan present in the materials [11,15].
- Comments 8: Line 86: Further specify the low-alcohol category.
Response 8: Thank you for your kind suggestion. It has been revised in the manuscript.
- Comments 9: Introduction: Better characterize the definition and market of craft beers.
Response 9: Thank you for your kind suggestion. It has been revised in the manuscript.
Lines 58-61: Craft beer is beer brewed using only four ingredients: malt, yeast, hops, and water. In addition, innovative ingredients, rather than cheap additives, are used as adjuncts to enhance flavor, resulting in a beer with unique taste profiles, diversity, and a sense of history or innovation.
- Comments 10: Materials and Methods: Check the adequacy of equations and ensure style standardization.
Response 10: Thank you for meticulously pointing this out. We have reconfirmed the adequacy of the formula and ensured that the format is standardized.
- Comments 11: Lines 188 and 190: Standardize units throughout the text (e.g., min, minutes, etc.).
Response 11: Thank you for pointing out this error. We have rewritten these sentences as follows.
Line 203: for 70 "min".
- Comments 12: Line 219: Briefly explain the procedure and any modifications made.
Response 12: Thank you for carefully pointing this out. The beer color determination was conducted as described in the EBC method, following Analytica EBC 9.6 without any modifications. It has been revised in the manuscript.
- Comments 13: Line 264: Insert the protocol number of the relevant ethics committee approval.
Response 13: We appreciate your valuable advice. In this manuscript, we did not conduct any animal experiments or invasive human trials. Furthermore, Pichia kluyveri, the non-Saccharomyces yeast used in this research for brewing applications, has GRAS (Generally Recognized As Safe) status. The low-alcohol beer developed for this research was brewed and bottled in the pilot plant of our campus and then provided to participants for sensory evaluation. We have previously used the same equipment to brew other beer products (photos of the production equipment and products are Appendix 1), all of which comply with hygiene and safety standards. We have also prepared an informed consent form (Appendix 2) to seek the consent of participants for the sensory evaluation experiment. During the sensory evaluation, participants visually assessed the beer’s appearance, detected its aroma through smell, and evaluated its bitterness and flavor through tasting.
- Comments 14: Lines 313-315: Provide a clearer explanation of microbial metabolism and the inhibitions that occur in co-culture.
Response 14: Thank you very much for your precious comment. The relevant information in the manuscript has been revised, and the details are as follows.
Lines 327-329: Canonico et al. [56] also reported that strains exhibiting lower fermentation kinetics in wort indicated that would be advantageous for low-alcohol beer production.
- Comments 15: Line 360: Further explain the term "attenuation."
Response 15: Thank you for the positive comments. The attenuation, expressed as a percentage, quantifies the amount of sugar yeast must ferment in beer wort and significantly influences the beer's body, sweetness, and flavor profile. The attenuation measured with a hydrometer is AA (based on observational measurements, not a physically precise quantity), meaning properly controlling the AA provides the beer with a specific flavor profile.
Source:
- HOME BREW answers:
https://homebrewanswers.com/yeast-attenuation/ - Everything You Need to Know about Attenuation:
https://beerandbrewing.com/everything-you-need-to-know-about-attenuation/
- Comments 16: Line 414: Clarify the classification of alcohol in relation to beer commercialization. Regardless of higher or lower concentration, products are classified as alcoholic beverages under the same legislation.
Response 16: Thank you for the positive comments. The specifications related to alcohol content have been explicitly supplemented in the issues above. The low alcohol content mentioned here is, as we noted in the manuscript, to develop craft beers with reduced alcohol content (compared to control groups or traditional methods) with satisfactory flavor indicators.
We have revised the manuscript with the following details.
Lines 129-131: Therefore, this study aimed to develop a craft beer with low alcohol content "(compared to the control group or traditional methods)" and rich aroma "via fermentation",
Lines 400-402: Therefore, this study aimed to develop a craft beer with a lower alcohol content (compared to the control group or traditional methods) while ensuring satisfactory performance indicators.
- Comments 17: Lines 484 and 489: Provide more explanations on color, including the reactions involved in color production in beers.
Response 17: Thank you for the positive comments. We have revised the manuscript with the following details.
Lines 523-526: The research demonstrates that color is pivotal in consumers' acceptance of foods, rendering it one of the most crucial attributes [67,68]. However, consumers typically prefer visually appealing beer colors, encompassing a broad spectrum ranging from light to dark hues, including gold, yellow, pale straw, amber, copper, and brown to black [68-70].
Lines 530-536: Interestingly, the color of beer comes mainly from the phenolic compounds in the malt or hops, which, apart from providing the beer with a variety of colors, also contribute to the protection of light-sensitive elements, which also facilitate the preservation of the beer [68,69]. Furthermore, the thermal treatment of the formulated ingredients led to primary color changes in the beer through the Maillard reaction and caramelization of sugars and amino acids [68,69]. It has been reported that polyphenol oxidation is also implicated in these color changes [69].
- Comments 18: No relevant discussion is presented under "Appearance color." Please expand this section.
Response 18: Thank you for the positive comments. We have revised the manuscript with the following details.
Lines 545-549: Therefore, these phenomena can be attributed to the subtle effects of different yeast strains despite the influence of material (malt or hops) and heat treatment during manufacturing on MR production and polyphenol oxidation [68,69]. However, it should be noted that in this study, the observed variations primarily stem from the utilization of distinct yeast strains
- Comments 19: Line 541: In this case, would the beer be classified as non-alcoholic? Explain this better and describe the relevant legislation.
Response 19: Thank you for the positive comments. Indeed, as responded to in the comments above, this study develops a craft beer with a lower alcohol content (compared to a control or conventional method) while ensuring satisfactory performance indicators. In addition, relevant legislation regarding alcohol content has been added to the "Introduction".
- Comments 20: Include an image of the produced beers.
Response 20: We appreciate your valuable advice. We have provided a graphical abstract that includes the beer product (Appendix 3).
Thanks again to the reviewer for the great guidance and insightful suggestions to improve the quality of this article, and the author would give the highest tribute.
Reviewer 2 Report
Comments and Suggestions for Authors
The authors investigated the potential for brewing low-alcohol beer by performing co-fermentation using different ratios of Pichia kluyveri. Additionally, co-fermentation with S. cerevisiae and P. kluyveri enabled the production of beer with an ABV level reduced to approximately 2.98%. To enhance the quality of the paper and effectively convey information to the readers, the authors are encouraged to address the following points.
Miner comments
1.Line52: “need”. It’s better to use “demand”.
2.Line54: “non-Saccharomyces”. Saccharomyces should be Italic.
3.Line 100: "PK" likely stands for Pichia kluyveri. Although P. kluyveri is mentioned in the abstract, the authors should restate "Pichia kluyveri" in full when first mentioned in the main text, as this is standard practice. Additionally, if "PK" is intended as the strain name, its use is acceptable; however, if it refers to the genus and species, the authors should write "P. kluyveri" in italicized form. Please review the entire manuscript, including the figures, and make all necessary corrections accordingly.
4.Lines 414-422: This paragraph is confusing. The authors have already selected the ratio of strains for co-fermentation and begun explaining Table 2, but then suddenly shift back to explaining Figure 1C. Please move this paragraph to a more relevant section, likely after the explanation of Figure 1B (Line 358).
5. Can the authors compare the advantages of using P. kluyveri over other yeast species? Since the authors have noted that there are existing publications on co-fermentation for brewing low-alcohol beer, it would be valuable to highlight the differences, advantages, and disadvantages of this study compared to previous publications. Discussing this point would help engage readers and emphasize the uniqueness of the research.
Author Response
Reviewer #2: The authors investigated the potential for brewing low-alcohol beer by performing co-fermentation using different ratios of Pichia kluyveri. Additionally, co-fermentation with S. cerevisiae and P. kluyveri enabled the production of beer with an ABV level reduced to approximately 2.98%. To enhance the quality of the paper and effectively convey information to the readers, the authors are encouraged to address the following points.
Response: Thank you very much for your precious comment. All revisions in the manuscript are shown in green.
Miner comments
- Comments 1: Line52: “need”. It’s better to use “demand”.
Response 1: Thank you for bringing the error to our attention. We have revised the manuscript and highlighted it in green (Line 51).
- Comments 2: Line54: “non-Saccharomyces”. Saccharomyces should be Italic.
Response 2: Thank you for bringing the error to our attention. We have revised the manuscript and highlighted it in green (Line 53).
- Comments 3: Line 100: "PK" likely stands for Pichia kluyveri. Although P. kluyveri is mentioned in the abstract, the authors should restate "Pichia kluyveri" in full when first mentioned in the main text, as this is standard practice. Additionally, if "PK" is intended as the strain name, its use is acceptable; however, if it refers to the genus and species, the authors should write "P. kluyveri" in italicized form. Please review the entire manuscript, including the figures, and make all necessary corrections accordingly.
Response 3: Thank you for bringing the error to our attention. The manuscript has revised the abbreviation PK to "P. kluyveri" to mitigate potential misinterpretations.
We have revised the manuscript and highlighted it in green.
- Comments 4: Lines 414-422: This paragraph is confusing. The authors have already selected the ratio of strains for co-fermentation and begun explaining Table 2, but then suddenly shift back to explaining Figure 1C. Please move this paragraph to a more relevant section, likely after the explanation of Figure 1B (Line 358).
Response 4: Thank you for bringing the error to our attention. The revised manuscript has been relocated to lines 394-402 and highlighted in green.
- Comments 5: Can the authors compare the advantages of using P. kluyveri over other yeast species? Since the authors have noted that there are existing publications on co-fermentation for brewing low-alcohol beer, it would be valuable to highlight the differences, advantages, and disadvantages of this study compared to previous publications. Discussing this point would help engage readers and emphasize the uniqueness of the research.
Response 5: Thank you very much for your precious comment. The P. kluyveri strain differs from other yeast species in its exclusive utilization of glucose as an energy source rather than sucrose and maltose, as reported in the results and discussed. This metabolic characteristic not only reduces alcohol production but also imparts an unfavorable flavor profile. Therefore, this study attempted to conduct co-fermentation with different ratios of strains to improve the poor flavor by retaining the advantages of lower alcohol content.
Thanks again to the reviewer for the great guidance and insightful suggestions to improve the quality of this article, and the author would give the highest tribute.
Round 2
Reviewer 1 Report
Comments and Suggestions for Authors
Authors have improved the quality of the work.
Author Response
Thanks again to the reviewer for the great guidance and insightful suggestions to improve the quality of this article, and the author would give the highest tribute.